# GIM: Learning Generalizable Image Matcher from Internet Videos

**Xuelun Shen**[1†]**, Zhipeng Cai**[2†*]**, Wei Yin**[3†]**, Matthias Müller**[2]**, Zijun Li**[1]**, Kaixuan Wang**[3]**, Xiaozhi Chen**[3]**, Cheng Wang**[1*]

[†] Equal Contribution, [*] Corresponding athuor (cwang@xmu.edu.cn, zhipeng.cai@intel.com)
[1] Fujian Key Laboratory of Sensing and Computing for Smart Cities,
  Xiamen University, 361005, P.R. China
[2] Intel Labs, [3] DJI Technology

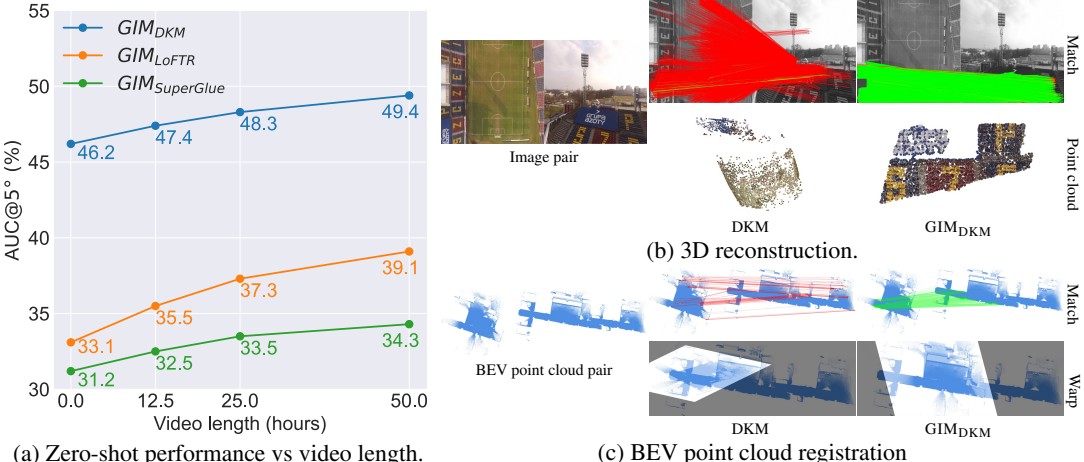

(a) Zero-shot performance vs video length.    (b) 3D reconstruction.

(c) BEV point cloud registration

Figure 1: **GIM overview.** We propose an effective framework for learning generalizable image matching (GIM) from videos. (a): GIM can be applied to various architectures and scales well as the amount of data (*i.e.*, internet videos) increases. (b): The improved performance transfers well to various downstream tasks such as 3D reconstruction. (c): Our best model GIM$_{DKM}$ also generalizes to challenging out-of-domain data like Bird Eye View (BEV) images of projected point clouds.

## Abstract

Image matching is a fundamental computer vision problem. While learning-based methods achieve state-of-the-art performance on existing benchmarks, they generalize poorly to in-the-wild images. Such methods typically need to train separate models for different scene types (*e.g.*, indoor vs. outdoor) and are impractical when the scene type is unknown in advance. One of the underlying problems is the limited scalability of existing data construction pipelines, which limits the diversity of standard image matching datasets. To address this problem, we propose GIM, a self-training framework for learning a *single generalizable* model based on any image matching architecture using internet videos, an abundant and diverse data source. Given an architecture, GIM first trains it on standard domain-specific datasets and then combines it with complementary matching methods to create dense labels on nearby frames of novel videos. These labels are filtered by robust fitting, and then enhanced by propagating them to distant frames. The final model is trained on propagated data with strong augmentations. Not relying on complex 3D reconstruction makes GIM much more efficient and less likely to fail than standard SfM-and-MVS based frameworks. We also propose ZEB, the first zero-shot evaluation benchmark for image matching. By mixing data from diverse domains, ZEB can thoroughly assess the cross-domain generalization performance of different methods. Experiments demonstrate the effectiveness and generality of GIM. Applying GIM consistently improves the zero-shot performance of 3 state-of-the-art image matching architectures as the number of downloaded videos increases (Fig. 1 (a)); with 50 hours of YouTube videos, the relative zero-shot performance improves by $6.9\% - 18.1\%$. GIM also enables generalization to extreme cross-domain data such as Bird Eye View (BEV) images of projected 3D point clouds (Fig. 1 (c)). More importantly, our single zero-shot model consistently outperforms domain-specific baselines when evaluated on downstream tasks inherent to their respective domains. The source code, a demo, and the benchmark are available at *https://xuelunshen.com/gim*.

# 1 INTRODUCTION

Image matching is a fundamental computer vision task, the backbone for many applications such as 3D reconstruction Ullman (1979), visual localization Sattler et al. (2018) and autonomous driving (Yurtsever et al., 2020).

Hand-crafted methods (Lowe, 2004; Bay et al., 2006) utilize predefined heuristics to compute and match features. Though widely adopted, these methods often yield limited matching recall and density on challenging scenarios, such as long-baselines and extreme weather. Learning-based methods have emerged as a promising alternative with a much higher accuracy (Sarlin et al., 2020; Sun et al., 2021) and matching density (Edstedt et al., 2023). However, due to the scarcity of diverse multi-view data with ground-truth correspondences, current approaches typically train separate indoor and outdoor models on ScanNet and MegaDepth respectively. Such domain-specific training limits their generalization to unseen scenarios, and makes them impractical for applications with unknown scene types. Moreover, existing data construction methods, which rely on RGBD scans (Dai et al., 2017) or Structure-from-Motion (SfM) + Multi-view Stereo (MVS) (Li & Snavely, 2018), have limited efficiency and applicability, making them ineffective for scaling up the data and model training.

To address these issues, we propose *GIM*, the first framework that can learn a single image matcher generalizable to in-the-wild data from different domains. Inspired by foundation models for computer vision (Radford et al., 2021; Ranftl et al., 2020; Kirillov et al., 2023), GIM achieves zero-shot generalization by self-training (Grandvalet & Bengio, 2004; Arazo et al., 2020) on diverse and large-scale visual data. We use internet videos as they are easy to obtain, diverse, and practically unlimited. Given any image matching architecture, GIM first trains it on standard domain-specific datasets (Li & Snavely, 2018; Dai et al., 2017). Then, the trained model is combined with multiple complementary image matching methods to generate candidate correspondences on nearby frames of downloaded videos. The final labels are generated by removing outlier correspondences using robust fitting, and propagating the correspondences to distant video frames. Strong data augmentations are applied when training the final generalizable model. Standard SfM and MVS based label generation pipelines (Li & Snavely, 2018) have limited efficiency and are prone to fail on in-the-wild videos (see Sec. 4.2 for details). Instead, GIM can efficiently generate reliable supervision signals on diverse internet videos and effectively improve the generalization of state-of-the-art models.

To thoroughly evaluate the generalization performance of different methods, we also construct the first zero-shot evaluation benchmark *ZEB*, consisting of data from 8 real-world and 4 simulated domains. The diverse cross-domain data allow ZEB to identify the in-the-wild generalization gap of existing domain-specific models. For example, we found that advanced hand-crafted methods (Arandjelović & Zisserman, 2012) perform better than recent learning-based methods Sarlin et al. (2020); Sun et al. (2021) on several domains of ZEB.

Experiments demonstrate the significance and generality of GIM. Using 50 hours of YouTube videos, GIM achieves a relative zero-shot performance improvement of $9.9\%$, $18.1\%$ and $6.9\%$ respectively for SuperGlue (Sarlin et al., 2020) LoFTR (Sun et al., 2021) and DKM (Edstedt et al., 2023). The performance improves consistently with the amount of video data (Fig. 1 (a)). Despite trained only on normal RGB images, our model generalizes well to extreme cross-domain data such as BEV images of projected 3D point clouds (Fig. 1 (c)). Besides image matching robustness, a *single* GIM model achieves cross-the-board performance improvements on various down-stream tasks such as visual localization, homography estimation and 3D reconstruction, even comparing to in-domain baselines on their specific domains. In summary, the contributions of this work include:

- GIM, the first framework that can learn a generalizable image matcher from internet videos.
- ZEB, the first zero-shot image matching evaluation benchmark.
- Experiments showing the effectiveness and generality of GIM for both image matching and various downstream tasks.

# 2 RELATED WORK

**Image matching methods:** Hand-crafted methods (Lowe, 2004; Bay et al., 2006; Rublee et al., 2011) use predefined heuristics to compute local features and perform matching. RootSIFT (Arandjelović & Zisserman, 2012) combined with the ratio test has achieved superior performance (Jin

et al., 2020). Though robust, hand-crafted methods only produce sparse key-point matches, which contain many outliers for challenging inputs such as low overlapping images. Many methods (Tian et al., 2017; Mishchuk et al., 2017; Dusmanu et al., 2019; Revaud et al., 2019; Tyszkiewicz et al., 2020) have been proposed recently to learn better single-image local features from data. Sarlin et al. (2020) pioneered the use of Transformers (Vaswani et al., 2017) with two images as the input and achieved significant performance improvement. The output density has also been significantly improved by state-of-the-art semi-dense (Sun et al., 2021) and dense matching methods (Edstedt et al., 2023). However, existing learning-based methods train indoor and outdoor models separately, making them generalize poorly on in-the-wild data. We find that RootSIFT performs better than recent learning-based methods (Sarlin et al., 2020; Sun et al., 2021) in many in-the-wild scenarios. We show that domain-specific training and evaluation are the cause of the poor robustness, and propose a novel framework GIM that can learn generalizable image matching from internet videos. Similar to GIM, SGP (Yang et al., 2021) also applied self-training (using RANSAC + SIFT). However, it was not designed to improve generalization and still trained models on domain-specific data. Empirical results (Sec. 4.2) show that simple robust fitting without further label enhancement cannot improve generalization effectively.

**Image matching datasets:** Existing image matching methods typically train separate models for indoor and outdoor scenes using MegaDepth (Li & Snavely, 2018) and ScanNet (Dai et al., 2017) respectively. These models are then evaluated on test data from the same domain. MegaDepth consists of 196 scenes reconstructed from 1 million internet photos using COLMAP (Schönberger & Frahm, 2016). The diversity is limited since most scenes are of famous tourist attractions and hence revolve around a central object. ScanNet consists of 1613 different scenes reconstructed from RGBD images using BundleFusion. ScanNet only covers indoor scenes in schools and it is difficult to use RGBD scans to obtain diverse images from different places of the world. In contrast, we propose to use internet videos, a virtually unlimited and diverse data source to complement the scenes not covered by existing datasets. The in-domain test data used in existing methods is also limited since they lack cross-domain data with diverse scene conditions, such as aerial photography, outdoor natural environments, weather variations, and seasonal changes. To address this problem and fully measure the generalization ability of a model, we propose ZEB, a novel zero-shot evaluation benchmark for image matching with diverse in-the-wild data.

**Zero-shot computer vision models:** Learning generalizable models has been an important research topic recently. CLIP (Radford et al., 2021) was trained on 400 million image-text pairs collected from the internet. This massive corpus provided strong supervision, enabling the model to learn a wide range of visual-textual concepts. Ranftl et al. (2020) mixed various existing depth estimation datasets and complementing them with frames and disparity labels from 3D movies. This allowed the depth estimation model to first time generalize across different environments. SAM (Kirillov et al., 2023) was trained on SA-1B containing over 1 billion masks from 11 million diverse images. This training data was collected using a "data engine", a three-stage process involving assisted-manual, semi-automatic, and fully automatic annotation with the model in the loop. A common approach for all these methods is to efficiently generate diverse and large scale training data. This work applies a similar idea to learn generalizable image matching. We propose GIM, a self-training framework to efficiently create supervision signals on diverse internet videos.

## 3 METHODOLOGY

Training image matching models requires multi-view images and ground-truth correspondences. Data diversity and scale have been the key towards generalizable models in other computer vision problems (Radford et al., 2021; Ranftl et al., 2020; Kirillov et al., 2023). Inspired by this observation, we propose *GIM* (Fig. 2), a self-training framework utilizing internet videos to learn a single generalizable model based on any image matching architecture.

Though other video sources are also applicable, GIM uses internet videos since they are naturally diverse and nearly infinite. To experiment with commonly accessible data, we download 50 hours (hundreds of hours available) of tourism videos with the Creative Commons License from YouTube, covering 26 countries, 43 cities, various lightning conditions, dynamic objects and scene types. See Appendix D for details.

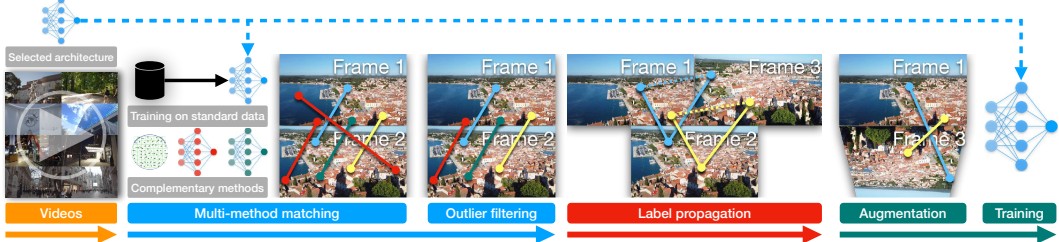

Figure 2: **GIM framework**. We start by downloading a large amount of internet videos. Then, given a selected architecture, we first train it on standard datasets, and generate correspondences between nearby frames by using the trained model with multiple complementary image matching methods. The self-training signal is then enhanced by 1) filtering outlier correspondences with robust fitting, 2) propagating correspondences to distant frames and 3) injecting strong data augmentations.

Standard image matching benchmarks are created by RGBD scans Dai et al. (2017) or COLMAP (SfM + MVS) (Schönberger & Frahm, 2016; Li & Snavely, 2018). RGBD scans require physical access to the scene, making it hard to obtain data from diverse environments. COLMAP is effective for landmark-type scenes with dense view coverage, however, it has limited efficiency and often fails on in-the-wild data with arbitrary motions. As a result, although millions of images are available in these datasets, the diversity is limited since thousands of images come from one (small) scene. In contrast, internet videos are not landmark-centric. A one hour tourism video typically covers a range of several kilometers (*e.g.*, a city), and has widely spread view points. As discussed later in Sec. 3.1, the temporal information in videos also allows us to enhance the supervision signal significantly.

## 3.1    SELF-TRAINING

A naive approach to learn from video data is to generate labels using the standard COLMAP-based pipeline (Li & Snavely, 2018); however, preliminary experiments show that it is inefficient and prone to fail on in-the-wild videos (see Sec. 4.2 for details). To better scale up video training, GIM relies on self-training (Grandvalet & Bengio, 2004), which first trains a model on standard labeled data and then utilizes the enhanced output (on videos) of the trained model to boost the generalization of the same architecture.

**Multi-method matching:**  Given an image matching architecture, GIM first trains it on standard (domain-specific) datasets (Li & Snavely, 2018; Dai et al., 2017), and uses the trained model as the 'base label generator'. As shown in Fig. 2, for each video, we uniformly sample images every 20 frames to reduce redundancy. For each frame $X$, we generate base correspondences between $\{X, X + 20\}$, $\{X, X + 40\}$ and $\{X, X + 80\}$. The base correspondences are generated by running robust fitting Barath et al. (2019) on the output of the base label generator. We fuse these labels with the outputs of different complementary matching methods to significantly enhance the label density. These methods can either be hand-crafted algorithms, or other architectures trained on standard datasets; see Sec. 4 for details.

**Label propagation:**  Existing image matching methods typically require strong supervision signals from images with small overlaps (Sarlin et al., 2020). However, this cannot be achieved by multi-method matching since the correspondences generated by existing methods are not reliable beyond an interval of 80 frames, even with state-of-the-art robust fitting algorithms for outlier filtering. An important benefit of learning from videos is that the dense correspondences between a video frame and different nearby frames often locate at common pixels. This allows us to propagate the correspondences to distant frames, which significantly enhances the supervision signal (see Sec. 4.2 for an analysis). Formally, we define $\mathbf{C}^{AB} \in \{0, 1\}^{r^A \times r^B}$ as the correspondence matrix of image $I^A$ and $I^B$, where $r^A$ and $r^B$ are the number of pixels in $I^A$ and $I^B$. A matrix element $c_{ij}^{AB} = 1$ means that pixel $i$ in $I_A$ has a corresponding pixel $j$ in $I^B$. Given the correspondences $\mathbf{C}^{AB}$ and $\mathbf{C}^{BC}$, to obtain the propagated correspondences $\mathbf{C}^{AC}$, for each $c_{ij}^{AB}$ in $\mathbf{C}^{AB}$ that is 1, if we can also find a $c_{j'k}^{BC} = 1$ in $\mathbf{C}^{BC}$, and the distance between $j$ and $j'$ in image $I^B$ is less than 1 pixel, we set $c_{ik}^{AC} = 1$ in $\mathbf{C}^{AC}$. Intuitively, this means that for pixel $j$ (or $j'$) in image $I^B$, it matches to both pixel $i$ in $I^A$ and pixel $k$ in $I^C$. Hence image $I^A$ and $I^C$ have a correspondence at location $(i, k)$.

To obtain strong supervision signals, we propagate the correspondences as far as possible as long as we have more than 1024 correspondences between two images. The propagation is executed on each sampled frame (with 20 frame interval) separately. After each propagation step, we double the frame interval for each image pair that has correspondences. As an example, initially we have base correspondences between every 20, 40 and 80 frames. After 1 round of propagation, we propagate the base correspondences from every 20 frames to every 40 frames and merge the propagated correspondences with the base ones. Now we have the merged correspondences for every 40 frames, we perform the same operation to generate the merged correspondences for every 80 frames. Since we have no base correspondence beyond 80 frames, the remaining propagation rounds do not perform the merging operation and keep doubling the frame interval until we do not have more than 1024 correspondences . The reason we enforce the minimum number of correspondences is to balance the difficulty of the learning problem, so that the model is not biased towards hard or easy samples. Though the standard approach of uniform sampling from different overlapping ratios (Sun et al., 2021) can also be applied, we find it more space and computation friendly to simply limit the number of correspondences and save the most distant image pairs as the final training data.

**Strong data augmentation:** To experiment with various existing architectures, we apply the same loss used for domain-specific training to train the final GIM model, but only calculate the loss on the pixels with correspondences. Empirically, we find that strong data augmentations on video data provide better supervision signals (see Sec. 4.2 for the effect). Specifically, for each pair of video frames, we perform random perspective transformations beyond standard augmentations used in existing methods. We conjecture that applying perspective transformation alleviates the problem where the camera model of two video frames is the same and the cameras are mostly positioned front-facing without too much "roll" rotation.

In practice, the major computation for generating video training data lies in running matching methods, and the average processing time per frame does not increase significantly w.r.t. the input video length. The efficiency and generality allows GIM to effectively scale up training on internet videos. It can process 12.5 hours of videos per day using 16 A100 GPUs, achieving a non-trivial performance boost for various state-of-the-art architectures.

## 3.2 ZEB: ZERO-SHOT EVALUATION BENCHMARK FOR IMAGE MATCHING

Existing image matching frameworks (Sarlin et al., 2020; Sun et al., 2021; Edstedt et al., 2023) typically train and evaluate models on the same in-domain dataset (MegaDepth (Li & Snavely, 2018) for outdoor models and ScanNet (Dai et al., 2017) for indoor models). To analyze the robustness of individual models on in-the-wild data, we construct a new evaluation benchmark *ZEB* by merging 8 real-world datasets and 4 simulated datasets with diverse image resolutions, scene conditions and view points (see Appendix E for details).

For each dataset, we sample approximately 3800 evaluation image pairs uniformly from 5 image overlap ratios (from 10% to 50%). These ratios are computed using ground truth poses and depth maps. The final ZEB benchmark thus contains 46K evaluation image pairs from various scenes and overlap ratios, which has a much larger diversity and scale comparing to the 1500 in-domain image pairs used in existing methods.

**Metrics:** Following the standard evaluation protocol (Edstedt et al., 2023), we report the AUC of the relative pose error within $5°$, where the pose error is the maximum between the rotation angular error and translation angular error. The relative poses are obtained by estimating the essential matrix using the output correspondences from an image matching method and RANSAC (Fischler & Bolles, 1981). Following the zero-shot computer vision literature (Ranftl et al., 2020; Yin et al., 2023), we also provide the average performance ranking across the twelve cross-domain datasets.

## 4 EXPERIMENTS

We first demonstrate in Sec. 4.1 the effectiveness of GIM on the basic image matching task — relative pose estimation. We evaluate different methods on both our zero-shot benchmark ZEB and the standard in-domain benchmarks (Sarlin et al., 2020). In Sec. 4.2, we validate our design choices with ablation studies. Finally, we apply the trained image matching models to various

Table 1: **Zero-shot matching performance.** GIM significantly improved the generalization of all 3 state-of-the-art architectures. IN means indoor model and OUT means outdoor model.

| Method | Mean Rank ↓ | Mean AUC@5°(%) ↑ | Real | | | | | | | | Simulate | | | |
|---|---|---|---|---|---|---|---|---|---|---|---|---|---|---|
| | | | GL3 | BLE | ETI | ETO | KIT | WEA | SEA | NIG | MUL | SCE | ICL | GTA |
| *Handcrafted* | | | | | | | | | | | | | | |
| ROOTSIFT | 7.1 | 31.8 | 43.5 | 33.6 | 49.9 | 48.7 | 35.2 | 21.4 | 44.1 | 14.7 | 33.4 | 7.6 | 14.8 | 35.1 |
| *Sparse Matching* | | | | | | | | | | | | | | |
| SUPERGLUE (IN) | 9.3 | 21.6 | 19.2 | 16.0 | 38.2 | 37.7 | 22.0 | 20.8 | 40.8 | 13.7 | 21.4 | 0.8 | 9.6 | 18.8 |
| SUPERGLUE (OUT) | 6.6 | 31.2 | 29.7 | 24.2 | 52.3 | 59.3 | 28.0 | 28.2 | 48.0 | 20.9 | 33.4 | 4.5 | 16.6 | 29.3 |
| GIM_SUPERGLUE | 5.9 | 34.3 | 43.2 | 34.2 | 58.7 | 61.0 | 29.0 | 28.3 | 48.4 | 18.8 | 34.8 | 2.8 | 15.4 | 36.5 |
| *Semi-dense Matching* | | | | | | | | | | | | | | |
| LoFTR (IN) | 9.6 | 10.7 | 5.6 | 5.1 | 11.8 | 7.5 | 17.2 | 6.4 | 9.7 | 3.5 | 22.4 | 1.3 | 14.9 | 23.4 |
| LoFTR (OUT) | 5.6 | 33.1 | 29.3 | 22.5 | 51.1 | 60.1 | 36.1 | 29.7 | 48.6 | 19.4 | 37.0 | 13.1 | 20.5 | 30.3 |
| GIM_LoFTR | 3.5 | 39.1 | 50.6 | 43.9 | 62.6 | 61.6 | 35.9 | 26.8 | 47.5 | 17.6 | 41.4 | 10.2 | 25.6 | 45.0 |
| *Dense Matching* | | | | | | | | | | | | | | |
| DKM (IN) | 2.6 | 46.2 | 44.4 | 37.0 | 65.7 | 73.3 | 40.2 | 32.8 | 51.0 | 23.1 | 54.7 | 33.0 | 43.6 | 55.7 |
| DKM (OUT) | 2.3 | 45.8 | 45.7 | 37.0 | 66.8 | 75.8 | 41.7 | 33.5 | 51.4 | 22.9 | 56.3 | 27.3 | 37.8 | 52.9 |
| GIM_DKM | 1.4 | 49.4 | 58.3 | 47.8 | 72.7 | 74.5 | 42.1 | 34.6 | 52.0 | 25.1 | 53.7 | 32.3 | 38.8 | 60.6 |
| GIM_DKM *with 100 hours of video* | | | | | | | | | | | | | | |
| GIM_DKM | | 51.2 | 63.3 | 53.0 | 73.9 | 76.7 | 43.4 | 34.6 | 52.5 | 24.5 | 56.6 | 32.2 | 42.5 | 61.6 |

downstream tasks (Sec. 4.3). To demonstrate the generality of GIM, we apply it to 3 state-of-the-art image matching architectures with varied output density, namely, SuperGlue (Sarlin et al., 2020), LoFTR (Sun et al., 2021) and DKM (Edstedt et al., 2023).

**Implementation details:** We take the official indoor and outdoor models of each architecture as the baselines. Note that the indoor model of DKM is trained on both indoor and outdoor data. For fair comparisons, we only allow the use of complementary methods that perform worse than the baseline during multi-method matching. Specifically, we use RootSIFT, RootSIFT+SuperGlue and RootSIFT+SuperGlue+LoFTR respectively to complement SuperGlue, LoFTR and DKM. We use the outdoor official model of each architecture as the base label generator in GIM. Unless otherwise stated, we use 50 hours of YouTube videos in all experiments, which provide roughly 180K pairs of training images. The GIM label generation on our videos takes 4 days on 16 A100 GPUs. To achieve the best in-domain and cross-domain performance with a single model, we train all GIM models from scratch using a mixture of original in-domain data and our video data (sampled with equal probabilities). The training code and hyper-parameters of GIM strictly follow the original repositories of the individual architectures.

## 4.1 MAIN RESULTS

**Zero-shot generalization:** We use the proposed ZEB benchmark to evaluate the zero-shot generalization performance. For all three architectures (Tab. 1), applying GIM produces a *single zero-shot* model with a significantly better performance compared to the best in-domain baseline. Specifically, the AUC improvement for SuperGlue, LoFTR and DKM is respectively $31.2 \rightarrow 34.3$, $33.1 \rightarrow 39.1$ and $46.2 \rightarrow 49.4$. GIM_SuperGlue performs even better than LoFTR (IN)/(OUT), despite using a less advanced architecture. Interestingly, the hand-crafted method RootSIFT (Arandjelović & Zisserman, 2012) performs better or on-par with the in-domain models on non-trivial number of ZEB subsets, *e.g.*, GL3, BLE, KIT and GTA. GIM successfully improved the performance on these subsets, resulting in a significantly better robustness across the board. Note that the performance of GIM did not saturate yet (Fig. 1), and further improvements can be achieved by simply downloading more internet videos. For example, using 100 hours of videos (Table 1, last row) we further improved the performance of GIM_DKM to $51.2\%$ AUC.

**Two-view geometry:** Qualitatively, GIM also provides much better two-view matching/reconstruction on challenging data. As shown in Fig. 3, the best in-domain baseline DKM (IN) failed to find correct matches on data with large view changes or small overlaps (both indoor and outdoor), resulting in erroneous reconstructed point clouds. Instead, GIM_DKM finds a large number of reliable correspondences and manages to reconstruct dense and accurate 3D point clouds. Interestingly, the robustness of GIM also allows it to be applied to inputs completely unseen during training. In Fig. 4, we apply GIM_DKM to Bird Eye View (BEV) images generated by projecting

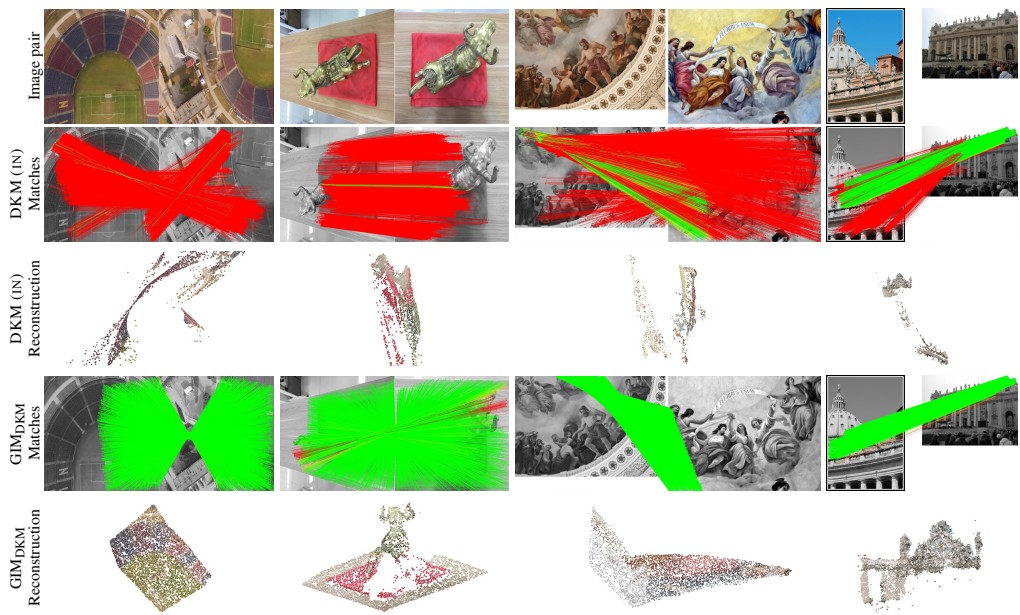

Figure 3: **Two-view reconstruction.** DKM returns many incorrect matches (red lines) on challenging scenes resulting in erroneous reconstruction. Applying GIM to the same architecture significantly improves matching and reconstruction quality.

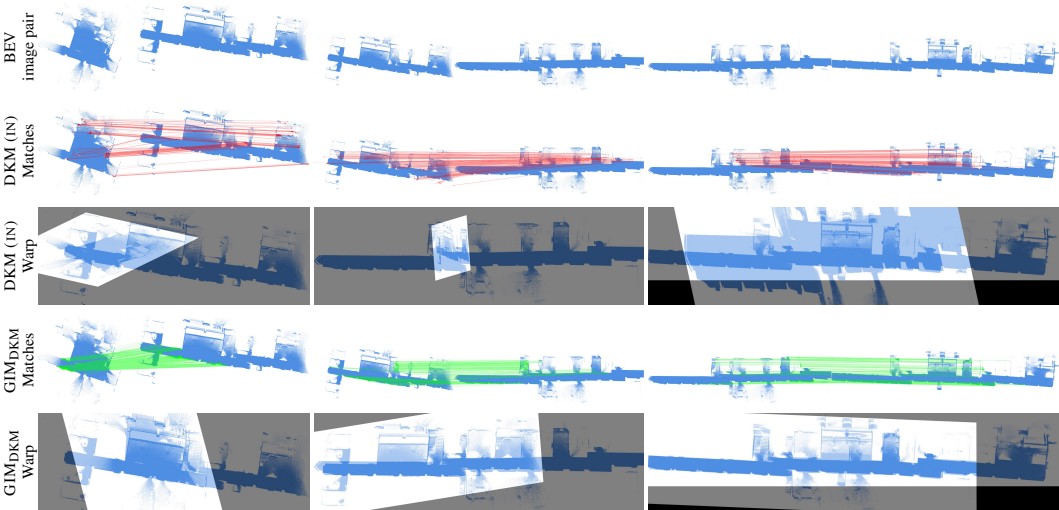

Figure 4: **Point cloud BEV image matching.** GIM$_{DKM}$ even successfully matches BEV images projected from point clouds despite never being trained for it.

the top-down view of two point clouds into 2D RGB images. The data comes from a real mapping application where we want to align the point clouds of different building levels in the same horizontal plane. Unlike the best baseline DKM (IN) that fails catastrophically, our model GIM$_{DKM}$ successfully registers all three pairs of point clouds even though BEV images of point clouds were never seen during training. Due to the space limit, we show the qualitative results for the other architectures in Appendix G.

**Multi-view Reconstruction:** GIM also performs well for multi-view reconstruction. To demonstrate the performance on in-the-wild data, we download internet videos for both indoor and outdoor scenes, extract roughly 200 frames for each video, and run COLMAP (Schönberger & Frahm, 2016) reconstruction but replace the SIFT matches with the ones from our experimented models. As shown

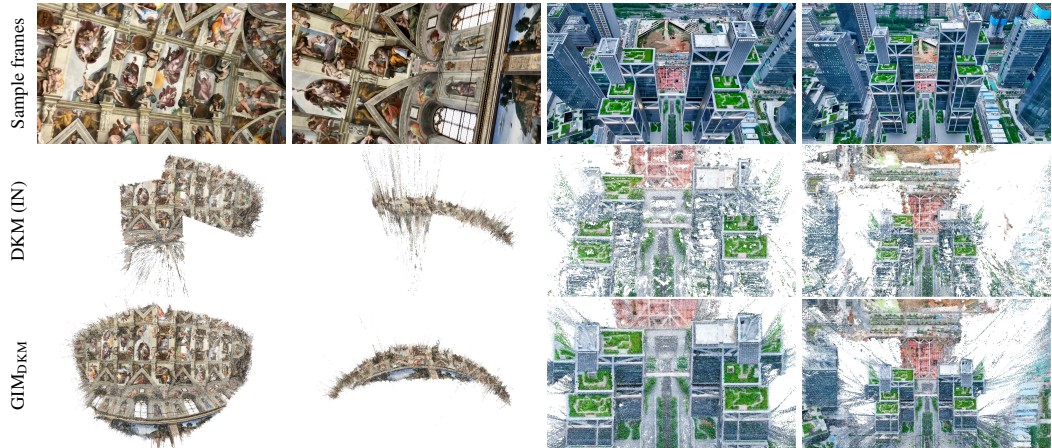

Figure 5: **Multi-view reconstruction.** GIM significantly improved the reconstruction coverage and accuracy.

Table 2: **Ablation study.**

| | Models | AUC@5°(%) |
|---|---|---|
| | GIM$_{\text{DKM}}$ | 49.4 |
| 1) | replace 50h video to 25h | 48.3 |
| 2) | replace 50h video to 12.5h | 47.4 |
| 3) | only use RootSIFT to generate labels | 49.3 |
| 4) | w/o data augmentation | 49.2 |
| 5) | w/o propagation | 47.1 |
| 6) | use COLMAP labels, same computation and time as 2) | 46.5 |
| 7) | w/o video | 46.2 |

Table 3: **Homography estimation.**

| Method AUC (%) → | @3px | @5px | @10px |
|---|---|---|---|
| SUPERGLUE (OUT) | 53.9 | 68.3 | 81.7 |
| GIM$_{\text{SuperGlue}}$ | **54.4** | **68.8** | **82.3** |
| LoFTR (OUT) | 65.9 | 75.6 | 84.6 |
| GIM$_{\text{LoFTR}}$ | **70.6** | **79.8** | **88.0** |
| DKM (OUT) | 71.3 | 80.6 | 88.5 |
| GIM$_{\text{DKM}}$ | **71.5** | **80.9** | **89.1** |

in Fig. 5, applying GIM allows DKM to reconstruct a much larger portion of the captured scene with denser and less noisy point clouds.

**In-domain performance:** We also evaluate different methods on the standard in-domain datasets. Due to the space limit, we report the result in Appendix F. Though the improvement is not as significant as in ZEB because individual baselines overfit well to the in-domain data, GIM still performs the best on average (over the indoor and outdoor scenes). This result also shows the importance of ZEB for measuring the generalization capability more accurately.

### 4.2 ABLATION STUDY

To analyze the effect of different GIM components, we perform an ablation study on our best-performing model GIM$_{\text{DKM}}$. As shown in row 1, 2 and 7 of Tab. 2, the performance of GIM consistently decreases with the reduction of the video data size. Meanwhile, adding only a small amount (12.5h) of videos already provides a reasonable improvement compared to the baseline (46.2% to 47.4%). This shows the importance of generating supervision signals on diverse videos. Using only RootSIFT Arandjelović & Zisserman (2012) to generate video labels, the performance of GIM reduces slightly. Comparing the performance between rows 3 and 1, we can see that generating labels on more diverse images is more important than having advanced base label generators. Removing label propagation reduces the performance more than lack of data augmentations and base label generation methods. Specifically, using 50 hours of videos without label propagation performs even worse than using the full GIM method on only 12.5 hours of videos.

We also experiment with the standard COLMAP-based label generation pipeline (Li & Snavely, 2018) (row 6). Specifically, we separate the downloaded videos into clips of 4000 frames and uniformly sample 200 frames for label generation. We apply the same GPU and time (roughly 1 day) as row 2 to run COLMAP SfM+MVS. COLMAP only manages to process 3.9 hours of videos, and fails to reconstruct 44.3% of them, resulting in only 2.2 hours of labeled videos (vs. 12.5 hours from GIM), and a low performance improvement of 46.2% to 46.5%.

Table 4: **Outdoor visual localization**. Unit: % of correctly localized queries (↑).

| Method | Day | Night |
| --- | --- | --- |
| | (0.25m,2°) / (0.5m,5°) / (1.0m,10°) | |
| SUPERGLUE (OUT) | 89.8 / 96.1 / 99.4 | 77.0 / **90.6** / **100.0** |
| GIM$_{\text{SUPERGLUE}}$ | **90.3** / **96.4** / 99.4 | **78.0** / 90.6 / **100.0** |
| LoFTR (OUT) | 88.7 / 95.6 / 99.0 | 78.5 / 90.6 / 99.0 |
| GIM$_{\text{LoFTR}}$ | **90.0** / **96.2** / **99.4** | **79.1** / **91.6** / **100.0** |
| DKM (OUT) | 84.8 / 92.7 / 97.1 | 70.2 / **90.1** / 97.4 |
| GIM$_{\text{DKM}}$ | **89.7** / **95.9** / **99.2** | **77.0** / 90.1 / **99.5** |

Table 5: **Indoor visual localization**. Unit: % of correctly localized queries (↑)

| Method | DUC1 | DUC2 |
| --- | --- | --- |
| | (0.25m,10°) / (0.5m,10°) / (1.0m,10°) | |
| SUPERGLUE (IN) | 49.0 / 68.7 / 80.8 | 53.4 / 77.1 / 82.4 |
| GIM$_{\text{SUPERGLUE}}$ | **53.5** / **76.8** / **86.9** | **61.8** / **85.5** / **87.8** |
| LoFTR (IN) | 47.5 / 72.2 / 84.8 | 54.2 / 74.8 / 85.5 |
| GIM$_{\text{LoFTR}}$ | **54.5** / **78.3** / **87.4** | **63.4** / **83.2** / **87.0** |
| DKM (IN) | 51.5 / 75.3 / 86.9 | 63.4 / 82.4 / 87.8 |
| GIM$_{\text{DKM}}$ | **57.1** / **78.8** / **88.4** | **70.2** / **91.6** / **92.4** |

## 4.3 APPLICATIONS

**Homography estimation:** As a classical down-stream application (Edstedt et al., 2023), we conduct experiments on homography estimation. We use the widely adopted HPatches dataset, which contains 52 outdoor sequences under significant illumination changes and 56 sequences that exhibit large variation in viewpoints. Following previous methods Dusmanu et al. (2019), we use OpenCV to compute the homography matrix with RANSAC after the matching procedure. Then, we compute the mean reprojection error of the four corners between the images warped with the estimated and the ground-truth homography as a correctness identifier. Finally, we report the area under the cumulative curve (AUC) of the corner error up to 3, 5, and 10 pixels. We take the numbers from the original paper for each baseline.

As illustrated in Tab. 3, the GIM models consistently outperform the baselines, even though the baselines are trained for outdoor scenes already. Among all architectures, GIM achieves the most pronounced improvement on LoFTR, achieving an absolute performance increase of 4.7%, 4.2%, and 3.4% in the three metrics.

**Visual localization:** Visual localization is another important down-stream task of image matching. The goal is to estimate the 6-DoF poses of an image with respect to a 3D scene model. Following standard approaches (Sun et al., 2021), we evaluate matching models on two tracks of the Long-Term Visual Localization benchmark, namely, the Aachen-Day-Night v1.1 dataset (Sattler et al., 2018) for outdoor scenes and the InLoc dataset (Taira et al., 2018) for indoor scenes. We use the standard localization pipeline HLoc (Balntas et al., 2017) with the matches extracted by corresponding models to perform visual localization. We take the numbers from the original paper for each baseline. Since DKM did not report the result on the outdoor case, we use the outdoor baseline to obtain the performance number.

With a *single* model, GIM consistently and significantly out-performs the domain-specific baselines for *both* indoor (Tab. 5) and outdoor (Tab. 4) scenes. For example, we improve the absolute pose accuracy of DKM by $> 5\%$ for the (0.25m, 2°) metric in both indoor and outdoor datasets. For indoor scenarios, GIM$_{\text{DKM}}$ reaches a remarkable performance of **57.1 / 78.8 / 88.4** on DUC1 and **70.2 / 91.6 / 92.4** on DUC2. These results show that without the need of domain-specific training, a single GIM model can be effectively deployed to different environments.

## 5 CONCLUSION

We have introduced a novel approach *GIM*, that leverages abundant internet videos to learn generalizable image matching. The key idea is to perform self-training, where we use the enhanced output of domain-specific models to train the same architecture, and improve generalization by consuming a large amount of diverse videos. We have also constructed a novel zero-shot benchmark *ZEB* that allows thorough evaluation of an image matching model in in-the-wild environments. We have successfully applied GIM to 3 state-of-the-art architectures. The performance improvement increases steadily with the video data size. The improved image matching performance also benefits various downstream tasks such as visual localization and 3D reconstruction. A single GIM model generalizes to applications from different domains.

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

## A    TRAINING DETAILS

Given 8 A100 GPUs, it takes about 4, 5, and 7.5 days to train $GIM_{SuperGlue}$, $GIM_{LoFTR}$, and $GIM_{DKM}$ respectively. For small memory GPUs, we have tried 4 RTX-3090. It takes 15 days to generate training signals on 50 hours of videos. And for training, it takes 9.5, 12, and 19 days to train $GIM_{SuperGlue}$, $GIM_{LoFTR}$, and $GIM_{DKM}$ respectively. Gradient accumulation is needed when using the RTX-3090 GPU to ensure the same training batch size as on the A100 GPU.

## B    PREVENT OVERFITTING

Preventing overfitting is crucial when training GIM with large-scale video data. Overfitting can occur if the training videos are not diverse enough and biased towards certain types of scenes. To address this issue, we employ an effective and implicit regularization technique in GIM by carefully selecting video data from diverse geo-locations and scene conditions. We focus on YouTube tourism videos and describe our search and filter strategies in Sec. 3 of the paper. Table 6 provides detailed statistics of the selected videos, showcasing the diversity of the training data.

Another factor that helps prevent overfitting is the continually growing amount and diversity of internet videos. As the training data expands, the risk of overfitting is effectively reduced. Additionally, we apply strong data augmentations during GIM training to make the model resistant to noise. The effectiveness of this implicit regularization approach is demonstrated by our experimental results in the Zero-shot Evaluation Benchmark (ZEB).

Although not used in our video selection process, evaluating the trained model on the ZEB benchmark can also serve as a verification tool to determine whether a set of candidate videos are potentially biased or noisy. By assessing the model's performance on the diverse and challenging scenarios present in the ZEB, we can gain insights into the generalization capability of GIM and identify any potential overfitting issues.

## C    HANDLING NOISY AND LOW-QUALITY DATA

When selecting internet videos for GIM, we recommend choosing videos with high resolution, clear images, long duration, and a small number of scene transitions. To ensure these criteria are met, we focus on tourism videos, which are generally 0.5-2 hours long and feature a person holding the camera and recording their travel, mostly without transitions. We also remove the first and the last 5 minutes of each video. This preprocessing step is performed because the beginning and the end of tourism videos often contain quick previews or summaries, which can include non-smooth scene transitions. These strategies helps to minimize the impact of noise and low-quality data on the training process.

GIM also exhibits some inherent robustness to transitions in the video. Drastic changes from one frame to another make it very difficult for matches to survive after robust fitting. Even if a few matches manage to survive, it is challenging for the label propagation to continue. This property of GIM helps to mitigate the effects of sudden scene changes and transitions in the training videos.

Furthermore, GIM incorporates strong data augmentations during training, which introduces noise to make the model more resistant to various sources of error in the input data. By exposing the model to a wide range of augmentations, such as random cropping, flipping, and color jittering, we can improve its robustness to noise and low-quality data that may be present in the internet videos.

Despite these measures, it is important to acknowledge that the quality of the training data can still have an impact on the performance of GIM. While our video selection strategy and the model's inherent robustness help to mitigate the effects of noise and low-quality data, it is essential to strive for high-quality and diverse training videos whenever possible to ensure the best possible performance of GIM.

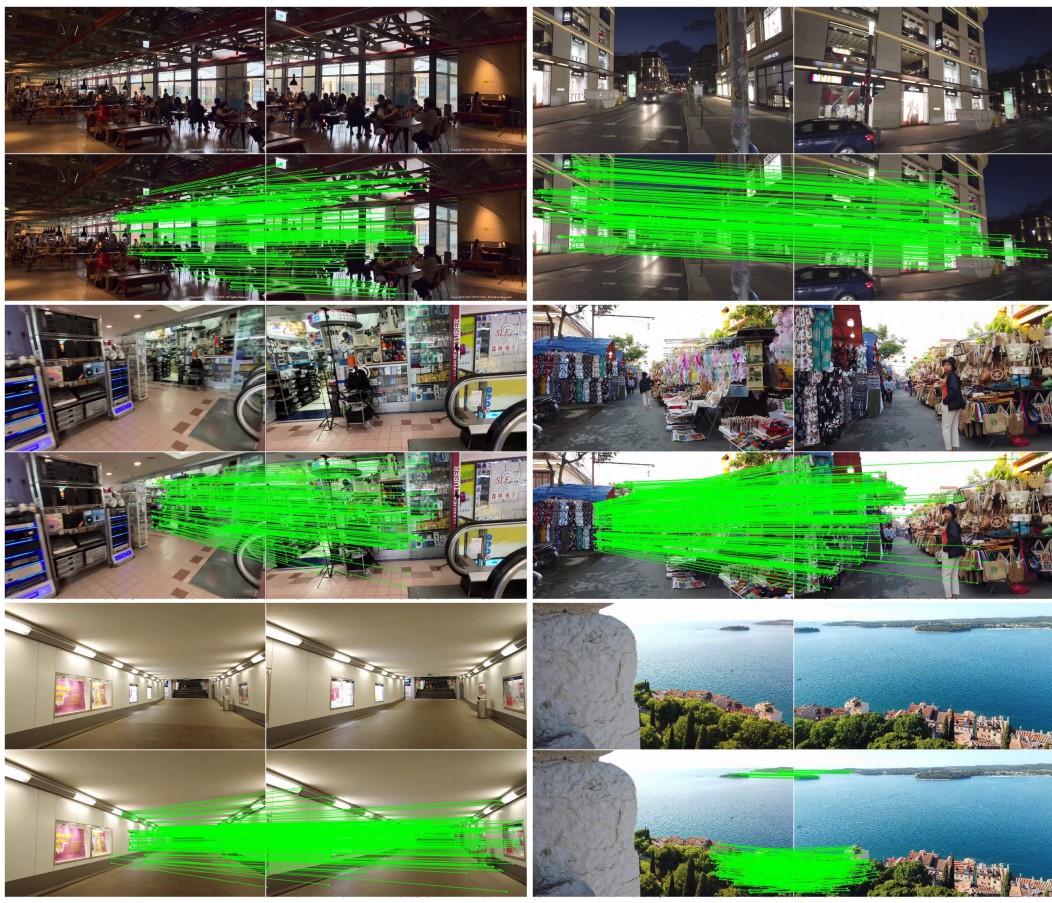

Figure 6: **Sample video data and the generated labels (for GIM_DKM).**

Table 6: **Video statistics.** The downloaded videos cover a wide range of scene types from 26 countries around the world, ensuring the diversity of the training data in GIM.

| Country (26) | City (43) | Scenario (39) |
|---|---|---|
| Italy, China, Korea, Bosnia, Greece, Poland, Turkey, Germany, Vietnam, Romania, Croatia, Austria, Albania, Hungary, America, Cambodia, Slovenia, Slovakia, Bulgaria, Thailand, Lithuania, Singapore, Montenegro, Herzegovina, Switzerland, Czech Republic | Rome, Seoul, Gdynia, Sopot, Gdask, Karpacz, Krakow, Trento, Tropea, Myslecinek, Bangkok, Santorini, Hoi An, Travnik, Singapore, Side, Brasov, Kampot, Bautzen, Ljubljana, Rovinj, Salzburg, Hue, Cottbus, Shkodr, Kemer, Bern, Prague, ilina, Budva, Debrecen, Vilnius, Kotor, Bodrum, Geneva, Varna, Shanghai, Milano, Dusseldorf, Busan, Los Angeles, Las Vegas, Irvine | Daytime, Driving, Suburbs, From Day to Night, Beach, Cave, Market, Sunny Day, Dock, Mountainous Area, Evening, Coast, Lights, Night, Park, Outskirts, Planetarium, Indoor and Outdoor Transition, Wilderness, Indoor, Storm rain, Lakeside, Chinatown, Street, Factory, Outdoor, Mountain Climbing, Mountain Road, City, Building, Shopping Mall, Small Town, Forest, Heavy Rain, Historic Building, Hollywood, Overcast Day, Historical Relics, Subway Station |

# D DETAILS OF VIDEO DATA

In this section, we show details of our video data. Tab. 6 shows the diverse geo-locations and scene types of our downloaded videos. Fig. 6 provides example training data (images and correspon-

dences) generated on video data, which covers both indoor and outdoor scenes, urban and natural environments, various illumination conditions.

# E  DETAILS OF ZEB

| Image Type | Dataset Name | Scenario | Image Size |
|---|---|---|---|
| Real Images | GL3D Shen et al. (2018) | aerial / wild | $1000 \times 1000$ |
| | BlendedMVS Yao et al. (2020) | objects | $1000 \times 1000$ |
| | ETH3D Indoor Schöps et al. (2017) | basement / corridor | $6000 \times 4136$ |
| | ETH3D Outdoor Schöps et al. (2017) | school / park | $6000 \times 4136$ |
| | KITTI Geiger et al. (2012) | driving | $1226 \times 370$ |
| | RobotcarWeather Maddern et al. (2017) | weather changes | $1280 \times 960$ |
| | RobotcarSeason Maddern et al. (2017) | seasonal changes | $1280 \times 960$ |
| | RobotcarNight Maddern et al. (2017) | sunlight changes | $1280 \times 960$ |
| Simulated Images | Multi-FoV Zhang et al. (2016) | driving | $640 \times 480$ |
| | SceneNet RGB-D McCormac et al. (2017) | living house | $320 \times 240$ |
| | ICL-NUIM Handa et al. (2014) | hotel / office | $640 \times 480$ |
| | GTA-SfM Wang & Shen (2020) | aerial / wild | $640 \times 480$ |

Table 7: **Datasets used to construct our zero-shot evaluation benchmark ZEB.** They contain varied image resolutions and scene conditions, with challenging view points (*e.g.*, aerial images). They also cover both real and simulated images.

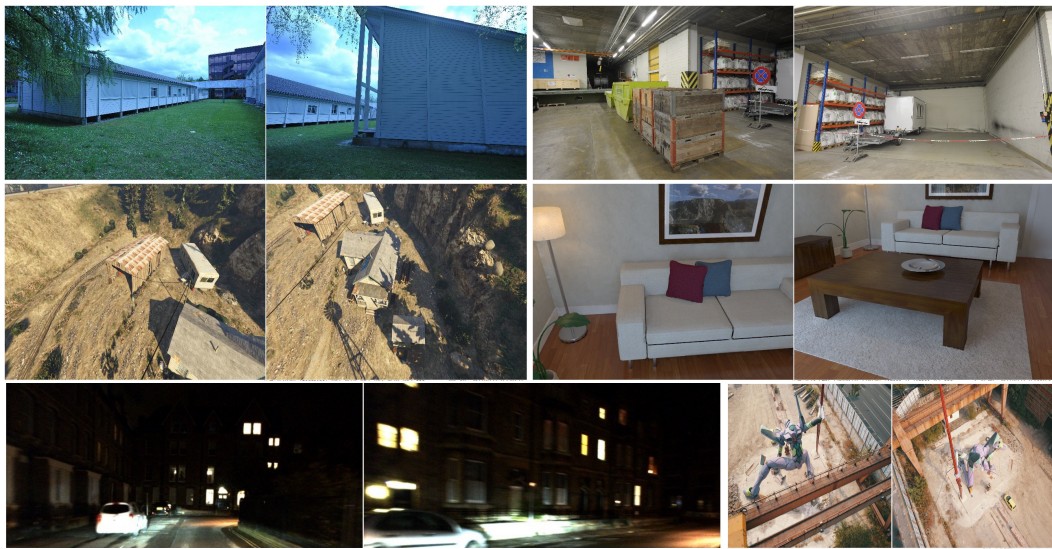

Figure 7: **Sample images of our zero-shot evaluation benchmark ZEB.** Various scene types, view points and lightning conditions are included to ensure a thorough evaluation of the matching robustness.

This section shows the details of the proposed ZEB benchmark. Specifically, Tab. 7 shows the 12 datasets used to construct ZEB, and the diverse scene conditions and images resolution covered by these datasets. We also show in Fig. 7 sampled image pairs in ZEB, covering varied scene types, view points and lightning conditions.

# F  IN-DOMAIN EVALUATION RESULT

As mentioned in Sec. 4.1, we also compared GIM with baselines on standard in-domain evaluation data, i.e., MegaDepth-1500 (Edstedt et al., 2023) and ScanNet-1500 (Edstedt et al., 2023). The

Table 8: **In-domain results** (↑). GIM still achieved the best overall performance on in-domain data.

| Method | AUC → | MegaDepth-1500 | | | ScanNet-1500 | | | Mean |
| | | @5° | @10° | @20° | @5° | @10° | @20° | |
| --- | --- | --- | --- | --- | --- | --- | --- | --- |
| SUPERGLUE (IN) | | 31.9 | 46.4 | 57.6 | **16.2** | **33.8** | **51.8** | 39.62 |
| SUPERGLUE (OUT) | | **42.2** | **61.2** | **76.0** | 15.5 | 32.9 | 49.9 | 46.28 |
| GIM$_{\text{SUPERGLUE}}$ | | 41.3 | 60.7 | 75.9 | 15.8 | 33.1 | 51.2 | **46.33** |
| LoFTR (IN) | | 4.0 | 9.3 | 18.4 | **22.1** | **40.8** | **57.6** | 25.37 |
| LoFTR (OUT) | | **52.8** | **69.2** | **81.2** | 18.0 | 34.6 | 50.5 | 51.05 |
| GIM$_{\text{LoFTR}}$ | | 51.3 | 68.5 | 81.1 | 19.5 | 37.3 | 55.1 | **52.13** |
| DKM (IN) | | 59.2 | 74.1 | 84.7 | **29.4** | **50.7** | **68.3** | 61.07 |
| DKM (OUT) | | 60.4 | 74.9 | 85.1 | 26.4 | 46.6 | 63.7 | 59.52 |
| GIM$_{\text{DKM}}$ | | **60.7** | **75.5** | **85.9** | 27.6 | 49.5 | 67.7 | **61.15** |

evaluation metric follows existing methods (Edstedt et al., 2023; Sun et al., 2021), and we take the numbers from the paper for each in-domain baseline. As shown in Tab. 8, though in-domain baselines already overfitted well on their trained domains, GIM still achieved the best average performance over indoor and outdoor scenes. The smaller performance gap comparing to the zero-shot scenario also shows the importance of the proposed ZEB benchmark, which can clearly reflect the generalization performance.

# G  FURTHER QUALITATIVE RESULTS

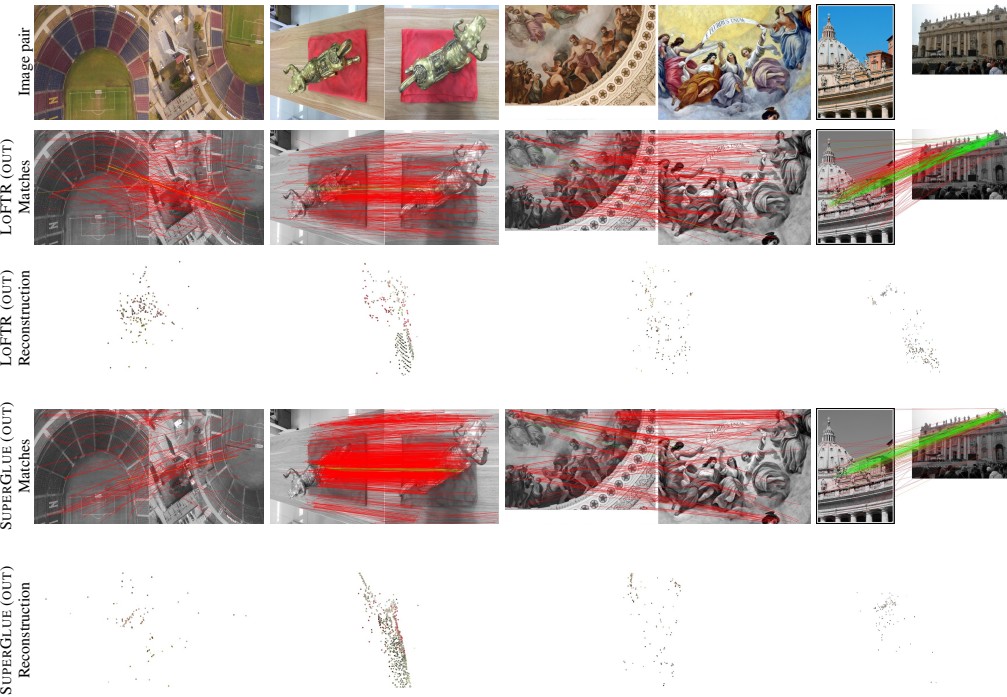

Figure 8: **Two-view reconstruction of other baselines.** We take the in-domain baseline that performs the best on ZEB for qualitative evaluation. Both LoFTR and SuperGlue generalize poorly on challenging in-the-wild data.

In Sec. 4.1, we only have space to show baseline results for the best architecture DKM. Here we provide the ones also for LoFTR and SuperGlue. Fig. 8 shows the two-view reconstruction results on in-the-wild images. Similar to DKM, the in-domain LoFTR and SuperGlue models also generalizes poorly on challenging in-the-wild data. Fig. 9 shows the results on BEV point cloud registration.

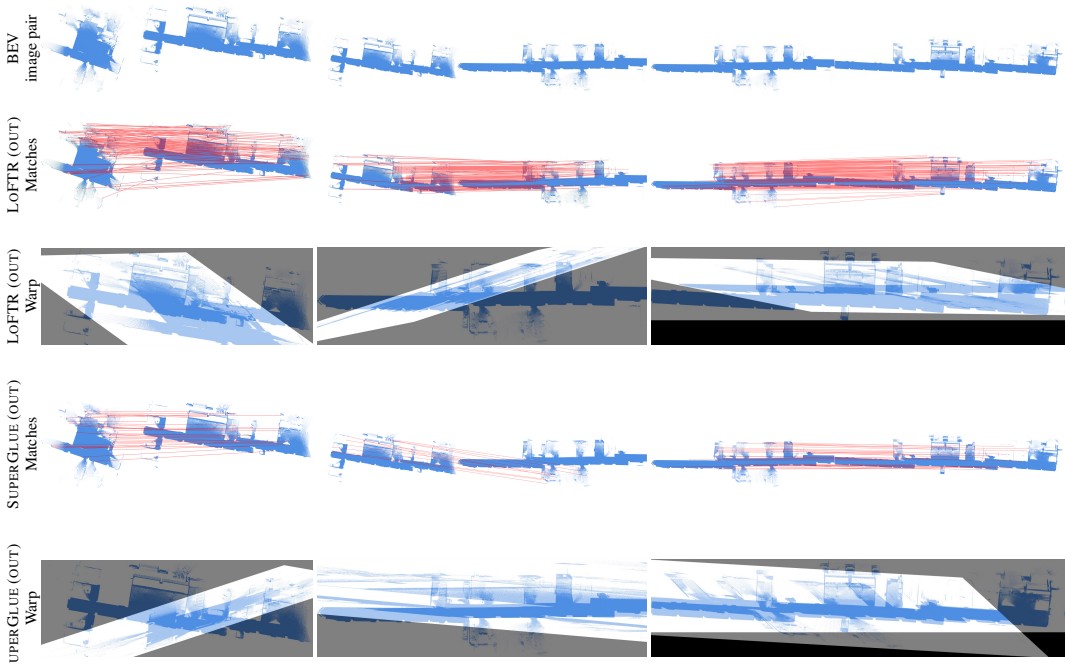

Figure 9: **Point cloud BEV image matching of other baselines.** The in-domain models of SuperGlue and LoFTR also failed to find reliable correspondences, resulting in wrong point cloud warping.

The in-domain LoFTR and SuperGlue models failed to find reliable matches and the correct relative transformations between two point clouds.

