# OpenReview forum: "GIM: Learning Generalizable Image Matcher From Internet Videos"
_ICLR.cc/2024/Conference — ICLR 2024 spotlight_

### Official Review · Reviewer_XccY · 2023-10-29

**Soundness:** 3 good
**Presentation:** 3 good
**Contribution:** 3 good
**Rating:** 6
**Confidence:** 3

**Summary:**

The paper introduces the GIM framework, aiming to address the challenge of generalizing image matching to diverse real-world scenarios. Taking cues from established computer vision models, GIM adopts a zero-shot generalization approach, leveraging the vast and varied nature of internet videos. The methodology involves a two-step training process: initial training on domain-specific datasets followed by an integration with several image matching techniques. This combined model seeks to identify potential correspondences in close video frames, with outlier detection mechanisms ensuring data quality. Notably, while traditional methods such as SfM and MVS have exhibited limitations in handling in-the-wild videos, GIM claims to efficiently provide reliable supervision signals for these scenarios, thereby promising to push the boundaries of current state-of-the-art models.

**Strengths:**

+ The introduction of the GIM framework is a significant advancement, marking the first attempt to leverage internet videos in training a universally applicable image matcher. This initiative promises to address generalization challenges across multiple real-world contexts.

+ The formulation of the ZEB benchmark is a noteworthy achievement. As a pioneering zero-shot evaluation benchmark integrating real-world and simulated data, ZEB is poised to become an instrumental tool in gauging the generalization capacities of existing models.

**Weaknesses:**

- The heavy reliance on internet videos for training might introduce biases or noise. The generalization capability of GIM, when trained on other diverse datasets, remains an unanswered question.

- With the consistent performance improvement with increased video data, there might be concerns about potential overfitting. Addressing this, perhaps with regularization techniques or other measures, would be crucial.

- Internet videos can be noisy, and their quality can vary. How resistant is GIM to such noise, and how does it handle low-quality data?

**Questions:**

-  While the use of internet videos is innovative, how did you ensure that the videos used for training represent a diverse range of scenarios, especially given the potential for internet content to have biases?

- How does the GIM model handle noise and varying quality within internet videos? Were any preprocessing steps or filters applied to ensure data quality?

- How did you address concerns of overfitting, especially with the consistent improvement seen with increasing video data?

---

> ### Author Response · Authors · 2023-11-21
> **Response to the reviews**
>
> **Weakness 1**: Thanks for your insightful comment. Though naively adding random internet videos can potentially introduce noise or bias, we have demonstrated in the experiments that by selecting tourism videos captured at diverse geo-locations and time, GIM can effectively generalize to various zero-shot inputs, e.g., images from different domains and BEV point clouds. As we discuss in the response to all reviewers, we have trained another model with 100h of data showing that GIM scales well with more video data.
>
> In terms of other diverse datasets, we focus on internet videos since they are arguably the most accessible and scalable data source. This ensures that the performance improvement of GIM can be reproduced by almost anyone on any future architecture. However, the key to the generalization of GIM is the data diversity and scale, rather than the source of the videos. Hence, we expect that as long as the dataset is **diverse** and **large**, it will work with GIM. So for example, adding proprietary videos captured in various environments, will likely further boost performance.
>
> We will clarify the importance of our video selection strategy and the key to the generalization of GIM in the camera-ready paper.
>
> **Weakness 2**: We completely agree that preventing overfitting is crucial. Specifically, overfitting can happen if the training videos are not diverse enough and biased towards certain types of scenes. An effective and implicit regularization that is applied in GIM is the selection of video data. We achieve this by selecting video data from diverse geo-locations and scene conditions; we focus on YouTube tourism videos and describe our search and filter strategies in Sec. 3 of the paper. Please refer to Table 6 of the paper for detailed statistics of the selected videos. Another potential factor in preventing overfitting is the amount of data. The continually growing amount and diversity of internet videos effectively reduces the risk of overfitting. We also added strong data augmentations during GIM training to make the model resistance to noise. Our experimental results in the zero-shot evaluation benchmark (ZEB) demonstrate the effectiveness of this implicit regularization approach.
>
> Finally, though never used in our video selection process, evaluating the trained model on the ZEB benchmark can also verify whether a set of candidate videos are potentially biased or noisy.
>
> We will include the above answers in camera-ready paper and hope they sufficiently address your concerns.
>
> **Weakness 3**: For the selection of internet videos, our recommendation is to choose videos with high resolution, clear images, long duration and small number of scene transitions. To ensure these points, we choose tourism videos. They are generally 0.5-2 hours long, with a person holding the camera and recording his/her travel, mostly without transitions. GIM also has some robustness to transitions in the video since drastic changes from one frame to another make it very difficult for matches to survive after robust fitting. Even if a few matches survive, it is very difficult for the label propagation to continue. Finally, GIM includes strong data augmentations during training, which introduces noise to make the model more resistant to various sources of error in the input data.
>
> We will include the above answer in camera-ready paper.
>
>
> **Question 1**: As mentioned in the answer of weakness 1, we did not randomly download videos from the internet. Instead, as mentioned in Sec. 3 of the main paper, we selectively used tourism videos for training. The selected tourism videos are usually captured by a person traveling around various places. Even only on YouTube, we can find tourism videos spanning thousands of hours and covering various regions, years, times, weather conditions, and lighting. These videos are not limited to a single type of scene; they include diverse activities like walking, cycling, driving and more. This ensures the data diversity and minimizes the bias.
>
> Based on your suggestion, we will describe our video selection process and its effectiveness in details in our camera-ready paper. The code will also be released upon acceptance.
>
> **Question 2**: Please refer to the details we have mentioned in weakness 3 for information how we handle noise and varying quality.
>
> Some pre-processing is indeed applied. First, we search YouTube for tourism videos using the keywords 'walk in' or 'walk through', and then downloaded videos that are at least 30 minutes long. Then we remove the first 5 minutes and the last 5 minutes of the video and keep only the middle part, because there may be a quick preview or summary which can contain non-smooth scene transitions. We will include these details in the appendix of camera-ready paper for clarity.
>
> **Question 3**: Please refer to the response of weakness 2.

---

### Official Review · Reviewer_jxyk · 2023-10-30

**Soundness:** 3 good
**Presentation:** 3 good
**Contribution:** 3 good
**Rating:** 8
**Confidence:** 2

**Summary:**

Observing that existing datasets for learning image-matching algorithm lacks diversity, the authors propose to learn image matchers from diverse tourism videos available on the internet. To obtain (pseudo-)labels for training, the authors propose to aggregate predictions from multiple matchers trained on small datasets and use a label propagation algorithm to propagate labels beyond nearby frames. Training on the pseudo-labeled data, existing image matching techniques demonstrate strong generalizability to unseen image domains, significantly outperforming image matchers trained on traditional datasets. In addition, the authors demonstrated that the proposed self-training framework showed strong scaling behavior, promising stronger image matchers for future work.

**Strengths:**

1. Simple and scalable framework: The proposed self-training framework is simple and scalable.
2. Strong zero-shot generalizability: Compared to image matchers trained on the traditional datasets, image matchers trained using self-training demonstrated stronger zero-shot generalizability, yielding more robust and performant image matchers.
3. Comprehensive experiments: Experiments include large collections of datasets and downstream tasks, showing the superiority of self-trained image matchers.

**Weaknesses:**

1. Lacking real indoor datasets in the benchmark: This is a nitpick but it would be great to have more real indoor datasets in the benchmark. Right now, most of the real datasets are driving-related and the indoor dataset only covers basements and corridors.

**Questions:**

Questions:
1. Will the performance continue to improve if the self-training is repeated multiple times or do the authors expect the models to start degrading due to noisy pseudo-labels?
2. Current work focuses on tourism videos. Would the approach work for other types of videos such as egocentric videos?


Suggestions:
1. (Related Work) Image Matching Datasets: MegaDepth is outdoor and ScanNet is indoor?
2. Section 3.1: “Multi-method Matching” was a little confusing. Would it make sense to use “Pseudo-matches Generation through Multi-method Matching”?
3. Table 1 KIT: GIM_LoFTR is slightly worse that LoFTR (out).
4. Table 2 w/o video: The reviewer assumes this is DKM (IN). It would be nice to specify this in the table.


Pre-rebuttal Rating: Overall, this is a good paper that presents an alternative to learning strong image matcher. The zero-shot investigation is insightful, and the framework’s strong zero-shot performance is encouraging despite its simplicity. The reviewer recommends accepting the paper prior to the rebuttal stage.

**Details Of Ethics Concerns:**

NA.

---

> ### Author Response · Authors · 2023-11-21
> **Response to the reviews**
>
> Thank you for the in-depth review. We will address all 4 suggestions in the camera-ready paper and provide responses to the major comments below.
>
> **Weakness 1.**: This is a very good suggestion. To facilitate better zero-shot evaluation, we will include more diverse real-world indoor scenes (e.g., SUN3D, ScanNet V2) during the code release of GIM and ZEB.
>
> **Question 1.**: Thank you for this insightful question. Since the ``iterative self-training process'' could be interpreted in different ways, we provide answers based on two possible interpretations. We will also clarify this in the camera-ready paper.
>
> **(Case 1) Using a fixed set of videos in multiple rounds of iterative self-training**: In this case, each round of self-training uses the current best GIM model (trained in the previous round) to re-generate the labels on the same set of videos, and then updates the GIM model with the re-generated labels. The expected improvement would come from better video labels and longer training time on the same set of videos. In terms of the video labels, Table 2 of the main paper shows that a better label generation method leads to better GIM performance. Hence, we would expect that this bootstrapping could in principle provide a further performance boost. However, the improvement of a single method may not be very significant in practice, due to the use of multiple complementary methods for label generation. In terms of the longer training time, image matching models are prone to overfitting, hence early stopping is applied during training. GIM does not change the baseline architecture or training curriculum and might also exhibit the overfitting problem for long training times, if the video dataset is too small and not expanded over time. Hence, if ``iterative self-training'' refers to case 1, it may slightly improve results but will probably not be as effective as expanding the video dataset at the same time. We discuss this case below and expect it to be the best way to boost the performance of GIM with iterative self-training.
>
> **(Case 2) Expanding the number of videos in multiple rounds of iterative self-training**: In this case, each round of self-training uses the current best GIM model (trained in the previous round) to generate labels on a **new** set of videos, and then performs the next round of self-training with both the old and new videos. Table 2 of the main paper shows that the most effective way to improve GIM is to add more video data, rather than generating better labels on a fixed set of videos. Hence, given the limited time and training resources (GPUs) and the potentially infinite amount of available internet videos, using the current best GIM model to generate labels on a new set of videos can best improve the performance. In this case, albeit less likely, overfitting may still happen when training the model for a long time. Hence, we recommend keeping early stopping active and re-initializing GIM weights for multiple rounds of self-training.
>
> **Question 2.**: We expect that GIM will also work with egocentric videos since there is nothing specific to the camera perspective in the algorithm. The more important aspect is that a similar level of diversity as in the case of the tourism videos is ensured (e.g., different geo-locations, lightning conditions, etc.). Training GIM on a mixture of diverse tourism videos and ego-centric videos may also be a good choice and boost generalization performance further. It is an interesting direction for further investigation.

---

> > ### Comment · Reviewer_jxyk · 2023-12-01
> > **Post Rebuttal**
> >
> > The reviewer has read the rebuttal and decided to keep the rating as is. One small suggestion: self-training/pseudo-labeling approaches are known to suffer from confirmation bias[1] so the reviewer suggests adding the discussion on iterative self-training (both case 1 and case 2) into the paper.
> >
> > [1] Arazo, Eric, Diego Ortego, Paul Albert, Noel E. O’Connor, and Kevin McGuinness. "Pseudo-labeling and confirmation bias in deep semi-supervised learning." In 2020 International Joint Conference on Neural Networks (IJCNN), pp. 1-8. IEEE, 2020.

---

### Official Review · Reviewer_UvwB · 2023-10-31

**Soundness:** 4 excellent
**Presentation:** 4 excellent
**Contribution:** 4 excellent
**Rating:** 10
**Confidence:** 5

**Summary:**

This paper proposed a Generalizable Image Matcher. It generalizes across training sets, view points, it works with BEV and even point cloud.

They also proposed ZEB, the first zero-shot evaluation benchmark for image matching. The claim it mixes data from diverse domains,
by using it one can assess the cross-domain generalization performance.

Extensive experiments on state-of-the-art baselines are compared.

**Strengths:**

The overview image in page.1 is impressive already. The method works on three strongest baseline (DKM, SuperGlue, and LoFTR) and improves them further more. It surprises me the method works with such huge view point differences and it also works with BEV pointcloud.

The training is using internet videos which prevents the COLMAP (SfM + MVS) bias for a single scene.

The proposed GIM is essentially a point matching ground-truth reinvention by using the label propagation through video with strong augmentation. However, it's so effective on every method by using the same training according to section 3.1.

I consider the simplicity not a weakness, but as a strength. If the proposed method is reproducable, I believe it would be the new standard of image matching.

The reconstruction results in Fig.3,4,5 are very impressive especially when you know it's training on sequence video instead of SfM alike scenario.

**Weaknesses:**

This paper is very impressive, I think the only thing left is just some implementation details becuase the self-training part is very short and only about the ground-truth instead of the training itself.

The only thing left is just open-sourcing the proposed code of the label propagation and training data to verify it's accuracy.

**Questions:**

Please share more about the training details. By domain specific training is it just swapping the GT or there are more details that's not being covered in the paper?

---

> ### Author Response · Authors · 2023-11-21
> **Response to the reviews**
>
> Thank you for the encouraging review, your appreciation of our simple yet effective method, and your valuable suggestions.
>
> **Weaknesses (Implementation details and code release)**: The training part mostly follows the baseline code as mentioned in the ``implementation details'' part of the paper.
>
> To ensure reproducibility, we will provide more details such as training time on different GPUs and for different base architectures, as discussed in the response to Reviewer-sN7h. We will also release the pre-trained GIM models, the training and ZEB evaluation data, and the code for label generation and model training.
>
> **Questions (Meaning of domain specific training)**: In the paper, we refer to training on standard image matching datasets (such as e.g., MegaDepth, ScanNet and unlike in-the-wild data from YouTube) as *domain-specific training*. In the experiments, we use the officially released models, i.e., SuperGlue (OUT), LoFTR (OUT) and DKM (OUT), as *domain-specific* models for GIM label generation. We will clarify this point in the camera-ready paper to avoid misunderstandings.

---

### Official Review · Reviewer_sN7h · 2023-11-01

**Soundness:** 3 good
**Presentation:** 3 good
**Contribution:** 3 good
**Rating:** 8
**Confidence:** 3

**Summary:**

This paper proposes a self-supervised method for image correspondence learning from easily accessible internet videos. The proposed method first trains an image-matching network on a standard dataset with GT supervision and then combines it with complementary image-matching methods to generate candidate correspondences between nearby frames of internet videos. Then, robust fitting is applied to remove outliers in the generated candidate correspondences. The remaining correspondences are used to re-train the image-matching network. By using this self-supervised training scheme, the performance improvement on existing image-matching models is impressive.

**Strengths:**

Many of the current deep networks suffer from poor generalization ability to unknown data distributions when the amount and diversity of training data are limited. Fine-tuning the model on the target data distribution with a small amount of data from the target domain with GT supervision is a natural way. However, obtaining GT information of the data from the target domain might not always be easy, especially in correspondence matching, pose estimation, 3D reconstruction, etc.

To address this challenge, this paper introduces a self-supervision strategy for image matchers using readily available internet videos. There is no need to run an SfM or COMAP pipeline, which is usually computationally expensive and un-robust to in-the-wild images, to obtain the GT correspondences. The performance improvement on three standard image-matching networks, diverse evaluation images, and various downstream tasks is impressive.

**Weaknesses:**

My comments below are more like questions instead of weaknesses.

(1) The proposed method combines a baseline image-matching network (e.g., SuperGlue, LoFTR, DKM) trained on a standard dataset and complementary image-matching methods to generate candidate correspondences. From the experiments section, the complementary image matching methods perform inferiorly than the baseline network. I have two questions here.
     a. Since the performance is inferior, why are they needed? Will this increase the number of estimated correspondences between two images and thus improve the performance?
     b. The introduction says "multiple" complementary image matching methods are used. While in the experiments section, it seems that only one complementary image matching method is used for each baseline method (SuperGlue, LoFTR & DKM). Will the number of complementary methods affect the performance?

(2) Does batch size affect the self-supervision performance? From the implementation details, the GIM label generation on 50 hours of YouTube videos takes 4 days on 16 A100 GPUs. How long will it take to re-train a baseline network using the generated labels (and on which type and how many GPUs)? Will the proposed method still work on an RTX 3090, which is commonly used in universities?

**Questions:**

Please refer to the weakness section above.

---

> ### Author Response · Authors · 2023-11-21
> **Response to the reviews**
>
> **Weakness (1)**: You are right, the complementary methods are introduced to increase the number of correspondences. Specifically multi-method matching is conducted only on nearby video frames ($<$80 frame interval), in which setting complementary methods are still able to provide reliable correspondences, albeit with inferior performance. The reason that complementary methods can increase the number of correspondences is that different methods often generate correspondences with different distributions. For example, RootSIFT generates correspondences on key point pixels. DKM outputs are mostly located on non-key point pixels. This strategy enhances the correspondence density between nearby video frames, which allows label propagation to produce training images with a larger pose difference.
>
> We do not just use 1 complementary method for each baseline, but rather all available methods that are inferior to the baseline, i.e., for training GIM$\_{SuperGlue}$, we use RootSIFT+SuperGlue to generate video labels. For training GIM$\_{LoFTR}$, we use RootSIFT+SuperGlue+LoFTR and for training GIM$\_{DKM}$, we use RootSIFT+SuperGlue+LoFTR+DKM to generate video labels. More complementary methods should in principle benefit the performance of GIM. For fair comparison, we use only inferior complementary methods so that we do not use labels generated by better methods to improve the baseline.
>
> We will clarify both points in camera-ready manuscript to avoid misunderstandings, thanks for your question.
>
> **Weakness (2)**: Yes, batch size affects the self-supervision performance; generally enlarging the batch size improves the performance. We follow the official implementation of the baselines to set the batch size.
>
> In terms of the training time of GIM, if we use 8 A100 GPUs, it takes about 4, 5 and 7.5 days to train GIM$\_{SuperGlue}$, GIM$\_{LoFTR}$, and GIM$\_{DKM}$ respectively.
>
> Yes, GIM works on GPUs with smaller GPU memory such as the RTX-3090, which is the GPU we used for initial experiments. For label generation, it takes 15 days to process 50 hours of videos on 4 RTX-3090 GPUs. For training, it takes 9.5, 12, and 19 days to train GIM$\_{SuperGlue}$, GIM$\_{LoFTR}$, and GIM$\_{DKM}$ respectively. We apply gradient accumulation when using the RTX-3090 GPU to ensure the same training batch size as on the A100 GPU.
>
> We will clarify these points in the appendix of the camera-ready manuscript to ensure the reproducibility of GIM across various setups.

---

> > ### Comment · Reviewer_sN7h · 2023-11-22
> >
> > Thanks to the authors for the convincing response. I keep my original rating for acceptance.

---

### Author Response · Authors · 2023-11-21
**Updated results with 100 hours of training videos**

We would like to thank all reviewers for recognizing the contribution of our work. After the submission deadline, we successfully trained GIM$\_{DKM}$ on 100 hours of videos and achieved a further performance boost.
At this time, we discovered that the cluster node used for the evaluation of the 50 hour GIM$\_{DKM}$ model has an I/O issue which leads to inaccurate results for the initial submission. We have fixed this issue and added further checks to the code to detect such issues and ensure the evaluation is robust and reproducible. Fortunately, the only model evaluated on this node was GIM$\_{DKM}$ (50h). We have carefully checked all reported results and confirm that all other numbers in the paper were correct and are reproducible. For reference, we report all reproduced DKM results, including the new GIM$\_{DKM}$ (100h), below and sincerely apologize for this error. We will also release the code, pre-trained models and data (both training and evaluation) upon acceptance.

| Method | Mean Rank↓ | Mean AUC@5° (%)↑ | GL3 | BLE | ETI | ETO | KIT | WEA | SEA | NIG | MUL | SCE | ICL | GTA |
| ------ | ---------- | ---------------- | --- | --- | --- | --- | --- | --- | --- | --- | --- | --- | --- | --- |
| DKM (in) | 3.5 | 46.2 | 44.4 | 37.0 | 65.7 | 73.3 | 40.2 | 32.8 | 51.0 | 23.1 | 54.7 | 33.0 | **43.6** | 55.7 |
| DKM (out) | 3.2 | 45.8 | 45.7 | 37.0 | 66.8 | **75.8** | 41.7 | 33.5 | 51.4 | 22.9 | **56.3** | 27.3 | 37.8 | 52.9 |
| $GIM\_{DKM}$ (50h) | 2.4 | 49.4 | 58.3 | 47.8 | 72.7 | 74.5 | 42.1 | 34.6 | **52.0** | 25.1 | 53.7 | 32.3 | 38.8 | 60.6 |
| $GIM\_{DKM}$ (100h) | **1.3** | **50.5** | **61.5** | **50.9** | **73.0** | 75.6 | **42.2** | **34.7** | **52.0** | **25.4** | 55.8 | **33.1** | 41.5 | **60.7** |

Though the performance of the 50 hour GIM$\_{DKM}$ model reduced from $50.1$ to $49.4$ AUC@5° after fixing the evaluation error, our 100 hour model significantly improves over the 50 hour model, reaching a new state-of-the-art performance of $50.5$ AUC@5°. Note that the 100 hour model performs better than the 50 hour model on **all** subsets of the ZEB evaluation data. The new performance figure ([click to show the anonymous figure](https://i.imgur.com/6YeVXQ6.png)) also clearly shows that the performance of GIM is not yet saturated even with 100 hours of video data, and can be further improved by adding even more training videos. The 100 hour result further supports the effectiveness and scalability of GIM.

---

### Meta-Review · Area_Chair_2FMn · 2023-12-15

**Metareview:**

All three reviewers give consistent and positive comments, and the major issues are: (1) Please share more about the training details in the final version. (2) Please show more details about the overfitting, and the effect of noisy data. After reading the comments, the AC recommends to accept this paper and encourages the authors to take the comments into consideration in their final version.

**Justification For Why Not Higher Score:**

Please see the detailed comments.

**Justification For Why Not Lower Score:**

All four reviewers give positive comments. The basic idea is novel and extensive experimental results demonstrate the effectiveness of this work.

---

### Decision · Program_Chairs · 2024-01-16

Accept (spotlight)